# Health insurance utilisation after ischaemic stroke in Sweden: a retrospective cohort study in a system of universal healthcare and social insurance

Carl Willers [ORCID],[1] Emma Westerlind,[2] Fredrik Borgström,[3,4] Mia von Euler,[5] Katharina S Sunnerhagen[2]

For numbered affiliations see end of article.

**Correspondence to**
Dr Carl Willers;
carl.willers@ki.se

## ABSTRACT

**Background** Stroke is one of the largest single-condition sources of the global burden of non-communicable disease in terms of disability-adjusted life-years and monetary costs, directly as well as indirectly in terms of informal care and productivity loss. The objective was to assess the population afflicted with ischaemic stroke in working age in the context of universal healthcare and social insurance; to estimate the levels of absence from work, the indirect costs related to that and to assess the associated patient characteristics.

**Methods** This was a retrospective register-based study; all individuals registered with an ischaemic stroke during 2008–2011 in seven Swedish regions, covering the largest cities as well as more rural areas, were included. Individual-level data were used to compute net days of sick leave and disability pension, indirect costs due to productivity loss and to perform regression analysis on net absence from work to assess the associated factors. Costs related to productivity loss were estimated using the human capital approach.

**Results** Women had significantly fewer net days of sick leave and disability pension than men after multivariable adjustment, and high-income groups had higher levels of sick leave than low-income groups. There were no significant differences for participants regarding educational level, region of birth or civil status. Indirect monetary costs amounted to €17 400 per stroke case during the first year, totalling approximately €169 million in Sweden.

**Conclusion** The individual's burden of stroke is heavy in terms of morbidity, and the related productivity loss for society is immense. Income-group differences point to a socioeconomic gradient in the utilisation of the Swedish social insurance.

## Strengths and limitations of this study

► The present study's underlying data are to a large extent extracted from population registries which implies very high coverage rates and hence a close-to-true picture of the actual population.
► Sick-leave costs for patients who had a stroke have been calculated before as well as after the stroke event, which adds important information in terms of incremental costs after stroke.
► Regression analysis of net days of absence included clinical as well as socioeconomic variables, implying broad adjustments and high possibility to understand the actual drivers.
► Informal care is costly, and no estimation of informal care has been included in this study due to lack of reliable data.
► The actual costs are in fact higher as any cases of sick benefits lasting shorter than 14 days are not registered in the Social Insurance Agency Database.

A non-negligible proportion of stroke patients are still in working age and these incur substantial direct and indirect costs. Stroke and ischaemic heart disease are the largest sources of the global burden of non-communicable disease in terms of disability-adjusted life-years.[1] Work disability after stroke is generally of more permanent nature compared with other conditions such as ischaemic heart disease.[3]

The Swedish social insurance covers individuals who live or work in Sweden; all people who are employed and whose employers pay social contributions to the agency are insured against loss of work-related salary up to a standardised ceiling.[4] The social insurance is considered a social security to inhabitants and companies, and also to society.[5] Better understanding of the situation of health

## BACKGROUND

Stroke is a leading cause of long-term disability globally and a resource-intensive disease, directly and indirectly.[1] Societal costs in terms of productivity loss are immense and have been for at least the last decade.[2]

insurance utilisation and its drivers could help policy-makers, governmental bodies and care providers with their work in designing guidelines, monitoring the development of care and optimising treatment. Evidence of the societal impact of stroke incidence, stroke care and recovery is needed to make good decisions on where to invest in the chain of care.

The aim with the present study was to map levels of health insurance utilisation in relation to ischaemic stroke—the year prior to and two years after—in a health system characterised by universal healthcare and social insurance. Furthermore, the aim was to estimate the distribution of productivity loss associated with stroke across age groups of working age, and to identify key drivers for utilisation of health insurance, for individuals afflicted with ischaemic stroke.

## METHODS
### Study population and data sources
Seven Swedish regions, covering approximately 65% of the Swedish working age population who had a stroke, contributed with data to the research database. The study population consisted of persons afflicted with ischaemic stroke during 2008–2011 and who were aged 18–63 years at the time of stroke. This to only include adults and to allow for two-year follow-up of health insurance utilisation, as the general retirement age in Sweden historically has been 65 years (64.6–64.7 during 2008–2011[6]). Participants were included independently of degree of sick leave or disability pension prior to stroke.

Stroke cases were identified based on registration of acute ischaemic stroke diagnosis (ICD-10 code I63) in regional administrative systems together with a registration in the Swedish Stroke Register. The regional administrative systems include information on health-care consumption and registered diagnoses. The Swedish Stroke Register is a national register for stroke-specific outcomes, reported on optional basis by the caregiver and with a coverage rate of 90% at the time, with aims to enable research and continuous improvement of stroke care to ensure high-quality and equitable care.[7] Follow-up amounted to at least two years for all participants. Relevant data from research registries were linked on individual level, including data on health insurance (Swedish Social Insurance Agency), socioeconomics and mortality (Statistics Sweden). Data sources are specified for each variable in table 1.

All data were pseudonymised.

### Variable definitions
Productivity loss was defined as the sum of sick leave and/or disability pension (early retirement to some degree) in monetary terms and based on net days (net productivity loss in terms of full working days; sick leave can formally be granted to an extent of 25%, 50%, 75% or 100% per day, and the individual is formally working the remaining proportion with a salary adjusted accordingly). Absence

was calculated based on available information from the Swedish Social Insurance Agency; sick leave lasting 14 days or shorter is financed by the employer and not available in the national registry. Hence, only sick leave periods lasting longer than 14 days were included in the analysis.

Information on prior stroke, history of atrial fibrillation and hypertension was based on administrative data, including diagnosis registration, for two years prior to the stroke. Information on dependency for activities of daily living (ADL) was based on data from the Swedish Stroke Register (in need of help with either dressing or toilet visits rendered an ADL value of 1, else 0), as was the computation of the modified Rankin Scale which was made in accordance with a published and validated method for approximation.[8] Information on living situation, income, educational level and origin was available from Statistics Sweden. Clinical information on level of consciousness and reperfusion treatment was extracted from the Swedish Stroke Register. Due to low coverage rate of the National Institutes of Health Stroke Severity (NIHSS) in the register (<50% complete for this study population), an approximation of stroke severity was captured via information on level of consciousness at hospital arrival (a version of the Reaction Level Scale which is used in the Swedish Stroke Register), which has been shown to be a good approximation of NIHSS for prediction of, for example, mortality.[9]

### Calculation of indirect costs
Indirect costs were calculated based on the human capital approach[10]; it was assumed that indirect cost after stroke for the individual participant was equivalent to the sum of income the participant would receive if not having a stroke (ie, based on the number of net days of absence and the level of income per day during the year prior to stroke), and the employer contribution that the employer pays for their employees (31.42% of the salary[11]): *[individual income per day]×[individual net days of work absence]×[1.3142]*. Indirect costs related to the first 14 days of the included cases were also estimated and included in the analysis. The authors, however, did not have any information whether the individual participant was in fact employed at the time of stroke. Information on disposable income was available on individual level and was used as explanatory variable in the regression analysis, then categorised into quartiles. The exchange rate for Swedish krona per euro (€) was set to 0.11198, which was the official exchange rate on 31 December 2011.[12]

Estimation of indirect costs related to productivity loss was made per age group (age spans set prior to the analysis: <35, 35–44, 45–54, 55–63 years of age at the time of stroke). The estimate was based on individual-level income data for the study population, available from Statistics Sweden.

**Table 1** Baseline characteristics of the study population, per age group

| | Age groups | | | | | |
|---|---|---|---|---|---|---|
| | <35 years | 35–44 years | 45–54 years | 55–63 years | | |
| n | 180 (3%) | 548 (8%) | 1 735 (26%) | 4 174 (63%) | 6637 | Data source |
| Sex, proportion women (%) | 52 | 42 | 37 | 33 | 35 | Statistics Sweden |
| ADL dependent (%) | 1.7 | 1.1 | 1.5 | 3.0 | 2.4 | Swedish Stroke Register |
| Prior stroke, last 2 years (%) | 3.4 | 4.0 | 4.3 | 4.6 | 4.4 | Regional administrative systems |
| Inpatient care the year prior to stroke (days) | 3.25 (SE 1.03) | 2.78 (SE 0.42) | 3.92 (SE 0.43) | 3.81 (SE 0.19) | 3.7 | Regional administrative systems |
| Level of consciousness (%) | | | | | | Swedish Stroke Register |
| Conscious | 89.8 | 94.0 | 93.4 | 93.9 | 93.7 | |
| Indolent | 5.7 | 5.2 | 5.2 | 4.9 | 5.0 | |
| Unconscious | 4.5 | 0.8 | 1.4 | 1.2 | 1.3 | |
| Atrial fibrillation, prior to stroke (%) | 1.1 | 2.9 | 5.4 | 10.9 | 8.5 | Regional administrative systems |
| Hypertension, prior to stroke (%) | 3.9 | 23.0 | 41.1 | 54.1 | 46.8 | Regional administrative systems |
| Reperfusion treatment (thrombolysis or thrombectomy, %) | 15.3 | 14.3 | 13.0 | 9.9 | 11.2 | Swedish Stroke Register |
| Single household (%) | 27.3 | 21.7 | 30.8 | 34.4 | 32.2 | Statistics Sweden |
| Disposable income (€, nominal amount) | | | | | | Statistics Sweden |
| Median | 17 217 | 24 468 | 21 646 | 20 935 | 21 311 | |
| IQR | 12 542 | 15 470 | 15 386 | 14 983 | 15 062 | |
| Educational level (%) | | | | | | Statistics Sweden |
| ≤9 years | 16.95 | 13.79 | 27.37 | 31.04 | 28.27 | |
| 10–12 years | 42.37 | 53.31 | 50 | 46.41 | 47.81 | |
| ≥12 years | 40.68 | 32.9 | 22.63 | 22.55 | 23.92 | |
| Born outside the EU (%) | 10.0 | 15.0 | 15.6 | 8.7 | 11.1 | Statistics Sweden |
| Sick leave the year prior to stroke (net days) | 17.7 | 19.1 | 18.4 | 17.3 | 17.7 | Swedish Social Insurance Agency |
| Disability pension the year prior to stroke (net days) | 25.0 | 34.5 | 74.0 | 108.7 | 91.2 | Swedish Social Insurance Agency |
| Total work absence the year prior to stroke (net days) | 42.7 (SE 8.1) | 53.6 (SE 4.9) | 92.4 (SE 3.5) | 126.0 (SE 2.1) | | Swedish Social Insurance Agency |

'ADL dependent' stands for dependency regarding activities of daily living, that is, need for support to manage day-to-day activities including hygiene and mobility. Level of consciousness at hospital arrival is subject to judgement by responsible clinician. Disposable income is equivalent to total income minus taxes. IQR denotes the interquartile range and entails the difference between quartile 3 and quartile 1 for each of the age groups. The categories of educational level stand for the Swedish equivalents of high school, secondary school and post-secondary education.
EU, European Union.

## Statistical analysis of factors associated with work absence

In addition, multivariable regression analysis was performed, adjusting for a broad range of patient characteristics of relevance (presented in table 2). Health insurance utilisation was analysed; net days of sick leave and disability pension (early retirement due to disability)

**Table 2** Association between patient characteristics and net days of absence (sick leave and/or disability pension); coefficients and 95% CI from regression analysis

| | Coefficient | 95% CI | P value |
|---|---|---|---|
| Sex, women | −0.085 | −0.167 to −0.004 | 0.040 |
| **Age** | | | |
| <35 | (Reference) | | |
| 35–44 | 0.144 | −0.114 to 0.402 | 0.274 |
| 45–54 | 0.158 | −0.081 to 0.397 | 0.196 |
| 55–63 | 0.150 | −0.085 to 0.386 | 0.211 |
| ADL dependent | 0.020 | −0.299 to 0.340 | 0.902 |
| Prior stroke (last 2 years) | −0.059 | −0.257 to 0.140 | 0.561 |
| Inpatient care the year prior to stroke | 0.001 | −0.002 to 0.004 | 0.562 |
| Net days of absence the year prior to stroke (sick leave and/or disability pension) | 0.003 | 0.003 to 0.003 | 0.000 |
| **Living situation** | | | |
| At home | (Reference) | | |
| At home with home care | −0.067 | −0.321 to 0.188 | 0.607 |
| At special housing | −0.067 | −0.436 to 0.303 | 0.724 |
| **Level of consciousness** | | | |
| Conscious | (Reference) | | |
| Indolent | 0.398 | 0.211 to 0.586 | 0.000 |
| Unconscious | 0.353 | −0.083 to 0.790 | 0.113 |
| Single household | 0.013 | −0.084 to 0.109 | 0.794 |
| **Disposable income** | | | |
| Quartile 1 | (Reference) | | |
| Quartile 2 | 0.309 | 0.201 to 0.417 | 0.000 |
| Quartile 3 | 0.417 | 0.302 to 0.532 | 0.000 |
| Quartile 4 | 0.218 | 0.097 to 0.340 | 0.000 |
| **Educational level** | | | |
| ≤9 years | (Reference) | | |
| 10–12 years | 0.022 | −0.068 to 0.112 | 0.632 |
| ≥12 years | −0.079 | −0.190 to 0.031 | 0.161 |

Continued

**Table 2** Continued

| | Coefficient | 95% CI | P value |
|---|---|---|---|
| Born outside of the EU | 0.033 | −0.092 to 0.157 | 0.609 |
| **Marital status** | | | |
| Married | (Reference) | | |
| Never married | 0.076 | −0.028 to 0.180 | 0.153 |
| Divorced | 0.030 | −0.075 to 0.134 | 0.575 |
| Widowed | −0.106 | −0.366 to 0.154 | 0.425 |

ADL, activities of daily living; EU, European Union.

were used as dependent variable, treated as a count variable, and analysed with negative binomial regression. The reason for treating it as a count variable was that the decision on the total amount of sick leave is iterative and based on a joint agreement between the doctor and the patient done during several occasions. Participants who died during follow-up were excluded from the analysis.

## RESULTS
### Study population
The study population amounted to 6637 individuals, whereof 35% were women. The study population was split into four age groups (table 1). The older the age group, the larger the volume of participants per age group; the youngest age group, <35 years, consisted of 180 individuals, and the oldest age group consisted of 4174 individuals. The younger the age group, the larger the proportion of women; 52% in the youngest age group, and 33% in the oldest age group. The prevalence of atrial fibrillation and hypertension seemed to increase with higher age, according to registration in medical records. The oldest age group (55–63 years at the time of stroke) versus the youngest (<35 years at the time of stroke) showed a prevalence of 10.9% vs 1.1% (atrial fibrillation) and 54.1% vs 3.9% (hypertension), respectively. Individual income (quartiles 1, 2 and 3) was highest in the group of participants aged 35–44 years at the time of stroke. The youngest age group was the most educated in terms of years spent in school with 40.7% having studied for 12 years or more.

### Calculation of work absence and indirect costs
The individual levels of absence from work due to sickness the year prior to stroke varied between the age groups (figure 1; CIs available in online supplemental table S1). The youngest age groups showed average absence of 42.7 and 53.6 net days, respectively, while individuals aged 45–54 years at the time of stroke presented with 92.4 net days of absence the year prior to stroke. Individuals aged 55–63 years at the time of stroke had 126.0 net days of absence the year prior to stroke. The

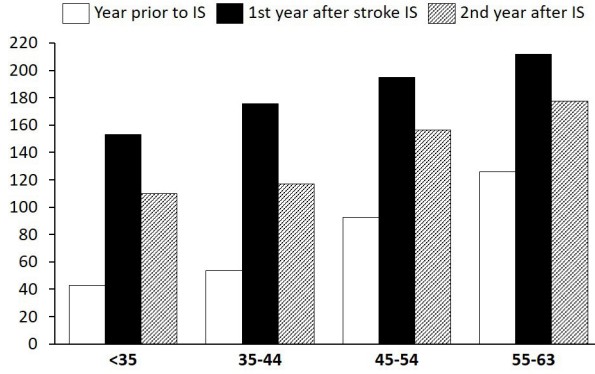

**Figure 1** Average health insurance levels used per individual, before and after stroke per age group for ischaemic stroke (IS): net days of sick leave and/or disability pension. The Y axis denotes number of net days of work absence, and the X axis denotes age groups expressed in years.

higher the age group, the higher the ratio of disability pension; the levels of net days prior to stroke were made up of 59% disability pension (<35 years), 64% disability pension (35–44 years), 80% disability pension (45–54 years) and 86% disability pension (55–63 years), respectively (the remaining proportion of net days regarded sickness benefit).

During the first year after stroke, levels of absence increased significantly from the year before; but overall, levels were quite similar across the different age groups: 153.1 (<35 years at stroke), 175.8 (35–44 years at stroke), 194.8 (45–54 years at stroke) and 211.8 (55–63 years at stroke) net days, respectively. The increase between the year prior to stroke versus after stroke was highest in the youngest age group (3.6 times higher) and lowest in the oldest age group (1.7 times higher).

Levels of absence the second year after stroke were significantly lower than levels the first year after stroke.

The largest relative drop in absence was found in the second youngest age group (35–44 years) and amounted to −33.5%. The age group with the lowest relative increase in the second year compared with the year prior to stroke was the oldest age group with 41% more net days.

The productivity loss amounted to €110 million during the first year after stroke (figure 2). The indirect costs were 27% lower during the second year after stroke and amounted to €76 million. The total age-group shares were similar during the first and second year although the second youngest age group was relatively less costly during the second year after stroke compared with the first year.

The average indirect cost per participant per year during the first and second year amounted to €11 600 and €7300 for the youngest age group, €16 600 and €9000 for participants aged 35–44 years, €16 800 and €11 500 for participants aged 45–54 years, and €18 000 and €13 200, respectively, for participants aged 55–63 years at the time of stroke. On average, indirect monetary costs amounted to €17 400 per stroke case on the first year and €12 200 on the second year.

### Statistical analysis of factors associated with work absence
When adjusting for clinical and sociodemographic factors potentially associated with levels of work absence after stroke, shown in table 2, belonging to a certain age group was not significantly associated with differences in absence from work due to sick leave or disability pension.

Women had significantly fewer net days of absence during the year after stroke compared with men (p=0.040), after multivariable adjustments. Participants in the lowest (first) income quartile had significantly lower absence levels compared with participants in the second (p=0.000), third (p=0.000) and fourth (p=0.000) quartiles. There were no significant differences for

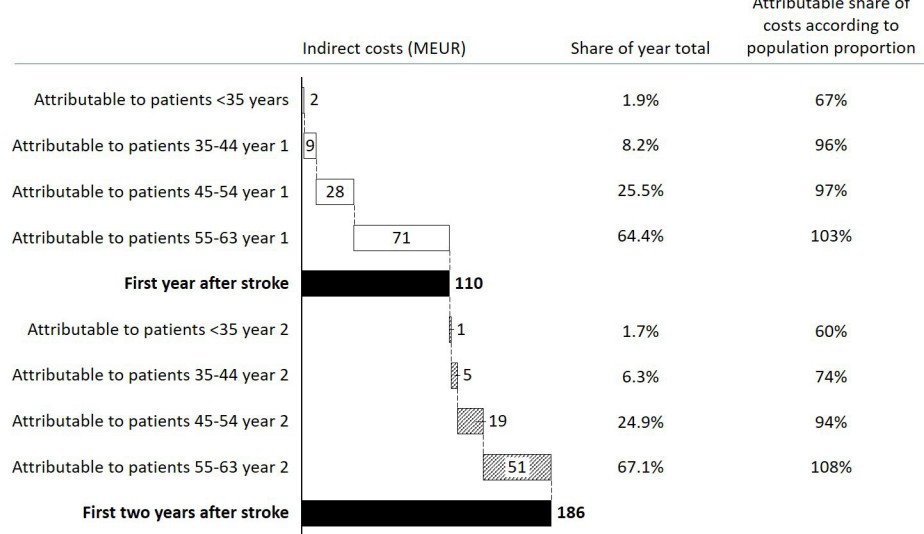

**Figure 2** Indirect costs in terms of productivity loss per age group years 1–2 after stroke, based on individual income data from Statistics Sweden. Proportion based on population contribution reflects whether the age-group share of total costs is representative or not (age group under-represented if <100% and over-represented if >100%). MEUR: million euros.

participants regarding educational level, region of birth or civil status. Performing the same regression analysis without previous health insurance utilisation (net days of work absence the year prior to stroke) as a covariate implied significant differences, with several sociodemographic covariates statistically significantly associated with net days of work absence after stroke (online supplemental table S2).

## DISCUSSION

The burden associated with stroke is immense—in terms of health-related quality of life[1] as well as money.[13–15] In addition to severe personal suffering, stroke affects society by putting a significant burden not only on direct healthcare resources but also on indirect, major and long-lasting productivity. The first-year indirect monetary costs for stroke in the present study totalled €110 million. Given that almost two-thirds of the Swedish stroke cases were the basis for analysis and that both urban and rural areas are represented, it is likely not unfair to make an extrapolation for all of Sweden; assuming equivalent incidence and equivalent levels of income for all of Sweden render an estimate of approximately €169 million annually, including only the first year after stroke. The costs are not limited to the first or second year (indirect monetary costs of €76 million for the study population, estimated to €117 million for the whole Swedish population) but expected to last several years after the stroke event.[16] Information on monetary productivity loss per year due to stroke in other countries is scarce, as shown in previous literature review,[17] but significantly lower levels than what was found in this study have been reported (eg, £2175 per case for the UK[18]). A study from Denmark, a context similar to the present study, estimated the indirect costs to €3100, although the majority of the subjects were no longer working (58.2% above 70 years of age).[19] Indirect costs related to stroke incidence in Sweden have been assessed in previous studies and found to be a minor proportion of the total cost, amounting to €1990–€5940 per stroke during the first year.[13–15] These studies did however include persons of all ages afflicted with stroke, and likely the indirect costs are significantly higher for the group in general working age as included in the present study. The estimate of total indirect costs of stroke in Sweden in this study was of the same order of magnitude as a 2020 published estimate of €122 million (productivity loss due to morbidity).[2]

Levels of sick leave the year prior to stroke were similar between the age groups (17.3–19.1 net days), while absence due to disability pension the year prior to stroke varied significantly (25.0–108.7 net days); the older the age group, the higher the absolute and relative levels of disability pension. The levels of net absence during the year prior to stroke point to the fact that the study's participants active on the labour market were more afflicted with disease than the average general population; pre-stroke levels of sick leave (absence excluding disability

pension) in the study population amounted to 17.7 net days (average of all age groups), while the national average for the same time period amounted to 6.0 net days (2010).[20] The patterns are the same looking at the health insurance as a whole; total absence in the study population the year prior to stroke amounted to more than three times the total absence in the general population for the study population (109.0 net days compared with 29.5 net days in 2010[20]). The proportion of women over the different age groups differed; the younger the age group, the higher the proportion of women. There are several reasons for this: hypertension and atrial fibrillation are less frequent in men than in women, while risk factors such as atrial fibrillation are of higher impact in women. In addition, pregnancy and contraceptives affect hormone levels and increase the risk of stroke.[21]

Sex and level of consciousness at hospital arrival were significantly associated with the level of absence the year after stroke; women were less absent on sick leave and/or disability pension as were participants who were conscious at hospital arrival compared with those who were indolent. The finding that female sex was significantly associated with lower levels of absence the year after stroke stood out compared with the general population. According to the Social Insurance Agency, women have higher risk of being on sick leave, a difference which has continued to increase since 2010.[22] Sick leave figures in public reports are often presented without multivariable adjustments which was the case in the present study. Furthermore, disposable income showed strong association with absence after stroke; participants in the lowest income group had significantly lower levels of absence compared with all other income groups.

Absence the year prior to stroke was positively correlated with absence the year after stroke. Interestingly, when this baseline characteristic was not taken into account, several socioeconomic indicators showed significant association with absence the year after stroke, including educational level, birth region and civil status.

There are several limitations with this study. Informal care is costly, and no estimation of informal care has been included in this study due to lack of reliable data. Furthermore, the actual costs measured are also in fact higher as any cases of sick benefits lasting shorter than 14 days are not registered in the Social Insurance Agency Database. There is also a lack of information whether the individuals were employed at the time of stroke or not, which would have been valuable to nuance the perspective on health insurance utilisation versus unemployment. The data may also be regarded as old as the last data point is almost 10 years old at the time of publication.

There are also several strengths with the present study. The underlying data are to a large extent extracted from population registries which implies very high coverage rates and hence a close-to-true picture of the actual population, and sick-leave costs for patients who had a stroke have been calculated before as well as after the stroke event, which adds important information in terms of

incremental costs after stroke. The possibility to adjust for several factors such as sex, age, comorbidity as well as several sociodemographic factors enables a better understanding of what patient characteristics are in fact associated with health insurance utilisation. The computation of productivity loss is based on objective data points for registered work absence and not related to any preference-based measures for health measures. Therefore, the findings regarding indirect costs related to stroke from the present study could be used together with preference-based measures and estimations of impact on health-related quality of life from other studies without implying risk for double counting, in line with previously published recommendations.[23]

During 2008, significant changes were implemented in the Swedish health insurance to limit the length of time possible to receive support for loss of income due to disability. These changes were implemented at the beginning of the time period assessed in the present study and could be expected to influence the estimation of costs equivalently over the time period studied.[24]

The fact that individuals with low income had fewer net days of sick leave is important to note as this points to a socioeconomic gradient in the utilisation of the general health insurance. As health equity is a topic treated in the Swedish healthcare act, it is particularly relevant to evaluate whether the distribution of governmental funding is actually equitable. Disposable income is an example of a factor that should not be associated with insurance utilisation after consideration has been taken to medical factors. There are several different factors that may impact and drive that gradient, but it is likely that factors such as higher general dependence on income in lower income professions and higher general knowledge regarding the social insurance system in higher income groups are of relevance to explain this gradient.[25] It is also likely that underlying themes such as knowledge of and ability to pursue primary and secondary stroke prevention are asymmetrically distributed across, for example, income groups. Such patterns reinforce already existing systematic differences in ill-health and actions to impact that are key to change the situation.

## CONCLUSIONS

Individuals who are afflicted with stroke in working age have a significantly higher pre-stroke level of health insurance (sick leave or disability pension) utilisation than the average in Sweden. The monetary burden of stroke is very high; the first-year indirect costs for an individual in working age amount to more than one-third of the gross domestic product per capita (reference year 2011). There is possibly a socioeconomic gradient in the utilisation of Swedish health insurance among patients who had a stroke; high-income individuals generally had higher levels of sick leave than did low-income individuals, while simultaneously adjusting for other relevant factors on patient level. Continuous assessments of the insurance utilisation among patients who had a stroke—as well as for other conditions—by governmental institutions are encouraged, to follow up on the equity in its design and practical implications.

**Author affiliations**
[1]Department of Neurobiology, Care Sciences and Society, Karolinska Institute, Stockholm, Sweden
[2]Department of Clinical Neuroscience, Institute of Neuroscience and Physiology, Sahlgrenska Academy, Goteborg, Sweden
[3]Department of Learning, Informatics, Management and Ethics, Medical Management Centre, Karolinska Institute, Stockholm, Sweden
[4]Quantify Research, Stockholm, Sweden
[5]School of Medicine, Örebro university, Örebro, Sweden

**Acknowledgements** The authors are grateful to the Swedish Stroke Register ( www.riksstroke.org) and the different healthcare regions for providing data for this study. They also wish to thank all patients, caregivers, reporting units and coordinators in Riksstroke as well as the Riksstroke Steering Committee.

**Contributors** CW and KSS conceived the idea for the study. CW performed the analyses and drafted the manuscript. CW, EW, FB, MvE and KSS contributed to the final version and approved the submitted version.

**Funding** This study was supported by grants from the Swedish Research Council (VR2017-00946), the Swedish Heart and Lung Foundation, the Swedish Brain Foundation, the Promobilia Foundation, the Norrbacka-Eugenia Foundation, as well as by grants from the Swedish state under an agreement between the Swedish government and the county councils, the ALF agreement (ALFGBG-718711).

**Competing interests** None declared.

**Patient and public involvement statement** It was not appropriate or possible to involve patients or the public in the design, or conduct, or reporting, or dissemination plans of this research.

**Patient consent for publication** Not required.

**Ethics approval** The study was approved by the Regional Ethical Review Board in Stockholm (2013/1541-31/5 and 2016/785-32).

**Provenance and peer review** Not commissioned; externally peer reviewed.

**Data availability statement** Data may be obtained from a third party and are not publicly available. The data set generated and analysed during the current study is not publicly available due to Swedish law; access to the data set was given to the researchers under the condition of timely valid ethical approval. Additional information regarding the availability is available from the corresponding author upon request.

**ORCID iD**
Carl Willers http://orcid.org/0000-0002-7616-9238

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
