## [Reviewer comments · BMJ Open]

ARTICLE DETAILS

TITLE (PROVISIONAL)	Health insurance utilization after ischemic stroke in Sweden: a retrospective cohort study in a system of universal healthcare and social insurance
AUTHORS	Willers, Carl; Westerlind, Emma; Borgström, Fredrik; von Euler, Mia; Sunnerhagen, Katharina

VERSION 1 – REVIEW

REVIEWER	Brystana G Kaufman Duke University
REVIEW RETURNED	05-Oct-2020

GENERAL COMMENTS	Using data from Sweden, the study describes the indirect costs of stroke, which is valuable in estimating cost-effectiveness and value of interventions to prevent and treat stroke. The data have strong generalizability due to the inclusion of 65% of the Swedish working-age stroke population. The analysis appears to be appropriate for the objectives. However, some of the methodological decisions need additional clarification/rationale, and the organization and consistency could be improved. Also, I could not evaluate the figures since they were hard to see. Abstract: Clarify what days are referred to in the results: "Women had significantly fewer net days..." This sentence sounds like results: The burden of stroke in terms of indirect monetary costs amounted to approximately 17.4 kEUR per stroke case first year (12.2 kEUR second year), totaling approximately 169 MEUR in Sweden first year (117 MEUR second year). I assume MEUR= million euros? Define at first use and use acronyms consistently. Conclusions: is the high cost of stroke in terms of productivity loss a key finding as well? Methods: Is there 100% success in linkage between data sources? If not, note how many were excluded at each step, including decedents? "sick leave can formally be granted to an extent of 25%, 50%, 75% or 100% per day" What does this mean, is this saying that individuals can receive a different % of salary? The formula for calculating indirect costs could be clarified by giving the formula. Were the adjusted days or raw days used to calculate productivity loss? The organization of the methods section could also help the reader by having 2 distinct sections: 1) calculation of indirect costs and 2) modeling outcomes, predictors, and analysis. Currently, it is not clear that these are independent analyses (assuming they are). "Participants deceased during follow-up were excluded from the
---

	analysis.” Does this mean anyone who died within 2 years? How many were excluded and how might this impact the estimation of indirect costs to society? Why did the authors choose to exclude NIHSS from the analysis rather than use multiple imputation approaches? Related, why was mRS not included in the models or descriptive tables? Please define specification and data source for all covariates in the methods (e.g. living situation, marital status, education etc are listed in the tables but not the methods text) Note the recommendations for calculations of productivity loss changed between the first and second panel on cost effectiveness; and this should be cited. “The authors did however not have any information whether the individual participant was in fact employed at the time of stroke.” I am confused by this since the intro states that only employed persons whose employers pay into the system can access the health insurance. Could the authors explain how unemployed persons would be included and how this would affect the analysis? Model choice seems appropriate for the outcome of “net days”, how were standard errors adjusted for clustering of outcomes by region? How was time accounted for in the model? Also, were two models run: one for year 1 and one for year 2? Results: Per stroke case results from the abstract does not appear in the results, and the discussion numbers do not match either results or abstract. Please improve consistency in which results are reported and confirm accuracy between abstract and manuscript. “The burden of stroke in terms of indirect monetary costs amounted to approximately 17.4 kEUR per stroke case first year (12.2 kEUR second year), totaling approximately 169 MEUR in Sweden first year (117 MEUR second year).” Discussion: “Assuming the equivalent incidence, and equivalent levels of income for the rest of Sweden renders an estimate of approximately 169 Million Euros annually – including only the first year after stroke.” This extrapolation requires some supporting data/rationale in the paper. Are these reasonable assumptions? E.g., if the included sample is limited to people who lived for 2 years does this generalize to all stroke patients nationally? Also, consider moving this finding to the results section, and discuss the assumptions and calculations for this analysis in the methods. How do the current estimates compare with prior estimates? Please expand this discussion and report actual indirect costs estimated from citations 13-15 so the reader can compare. Also, would be helpful to state the direct costs of stroke as cited, for the reader to compare. The bulleted strengths and limitations of the paper should be included in the discussion text. Since the conclusion highlights the socio economic gradient, it would be helpful to expand the paragraph explaining finding and potential drivers. Equity is mentioned in the last sentence of the conclusion, and it would be helpful to explain concerns about equity earlier in the discussion. Tables: Disability pension the year prior to stroke (net days) – is the difference between age group is reflecting the larger denominator? If this is the case, consider presenting a per capita rate? The other “net day” items like sick days and inpatient care are similar across age groups, so just wanted to check this one. Living situation – how is this variable defined, is this after or before stroke? Also, why do
--	---

	some variables in Table 2 not appear in Table 1? Why are there 3 categories for income in table 1 but 4 in table 2? Figures: something was wrong with the formatting and I could not evaluate these. The axes and labels were not visible on a black background?
--	--

REVIEWER	Bernhard Rauch IHF - Institute für Herzinfarktforschung Ludwigshafen, Germany
REVIEW RETURNED	24-Oct-2020

GENERAL COMMENTS	Ischemic stroke and health insurance utilization in a system of universal healthcare and social insurance. Journal: BMJ Open, Manuscript ID bmjopen-2020-043826 Article Type: Original research The aim of the present study was to assess the population in working age being afflicted with ischemic stroke in a system of universal healthcare and social insurance. By this way the absence from work and related indirect costs were evaluated within the context of the patient's characteristics. Based on registry data collected in seven Swedish regions between 2008 and 2011 net days of sick leave and disability, indirect costs due to productivity loss and associated factors potentially affecting these social consequences have been assessed. Compared to men women had significantly fewer net days of sick leave and disability, whereas the high-income group had higher levels of sick leave than low-income groups. There were no differences with regard to educational level, region of birth or civil status. The indirect costs amounted to approximately 17.400 EUR per stroke case in the first year and 12.200 EUR in the second year. Furthermore the data indicate a socioeconomic gradient in the utilization of the Swedish social insurance. Informal care has not been evaluated in this study due to lack of data. General Comments:  - The data sources are somewhat dated, thereby potentially not reflecting the actual situation in Sweden, especially taking into account changes in the Swedish health insurance after the time period under investigation. - The methodological approach, especially with respect to data acquisition and interactions between different data sources, are not clearly reported (see below). - The presented results have not been compared with data from other industrial countries, which would be of high interest. I also would expect a more detailed comparison of the presented data with other recent publications in this field. - The potential role of primary and secondary prevention has not been discussed. Specific Comments:  - Data sources and their linkage need to be described in more detail (e.g. Swedish stroke registry, Swedish Social Insurance Agency and socioeconomics and mortality by Sweden statistics). Which data have been acquired from which source? At best a flow chart should be delivered to better understand data sources and acquisition. The Swedish stroke registry needs to be described in detail (primary aim, population, time period, data acquisition and general outcomes). - Baseline characteristics in Table 1 are incomplete with respect to the medical situation (e.g. a complete spectrum of the most
---

	important comorbidities, risk diseases apart from atrial fibrillation and hypertension, severity of stroke, medication etc.) and also should include the total numbers of patients under investigation. At best one additional column should deliver the data from the whole population. The levels of consciousness should be defined and definitions should be given at the bottom line of the table. Also other terms like “disposable income” should be defined within the bottom line. Defining the educational level by “years of education” seems to be very vague and does not meet common standards.  - Drivers of post-stroke health insurance utilization in Table 2 are not readily comprehensible for the reader and need to be explained at a bottom line or a table legend. - The Figures all do have a black background and included text passages, numbers or symbols are not readable. This may be the result of a printing problems, but must be resolved. - Results: The text passage “The loss of productivity, calculated based on individual participants’ income together with additional employer contributions, amounted to 110 Million Euros during the first year after stroke (Figure 2)” is unclear. Do the 110 Million Euros refer to the whole population under investigation? How do these data compare with other European countries? - Results, page 10, lines 26-29: “When adjusting for other relevant factors of potential association with levels of absence after stroke, belonging to a certain age group was not significantly associated to differences in absence from work due to sick leave or disability pension”. Which “other relevant factors” have been included into the calculations? - Page 14, line 6: “During 2008, significant changes were implemented in the Swedish health insurance to limit the length of time possible to receive support for loss of income due to disability”. It may be regarded as a relevant limitation of the study by evaluating a dated study population (see general comment above). - Page 15, line 24: “Individuals that are afflicted with stroke in working age have a significantly higher pre-stroke level of health insurance utilization” This comment underscores the relevance of risk factors and comorbidities of stroke patients, potentially being present decades before incident stroke. Primary prevention therefore is essential for the individual but also for the community supporting the individual. This has not sufficiently been addressed neither within the study nor within the manuscript.
--	--

VERSION 1 – AUTHOR RESPONSE

Reviewer: 1

Comments to the Author

Using data from Sweden, the study describes the indirect costs of stroke, which is valuable in estimating cost-effectiveness and value of interventions to prevent and treat stroke. The data have strong generalizability due to the inclusion of 65% of the Swedish working-age stroke population. The analysis appears to be appropriate for the objectives. However, some of the methodological decisions need additional clarification/rationale, and the organization and consistency could be improved. Also, I could not evaluate the figures since they were hard to see.

Abstract:

Clarify what days are referred to in the results: “Women had significantly fewer net days...”

This sentence has now been updated to clarify that the results refer to the sum of net days of sick leave and disability pension.

This sentence sounds like results: The burden of stroke in terms of indirect monetary costs amounted to approximately 17.4 kEUR per stroke case first year (12.2 kEUR second year), totaling approximately 169 MEUR in Sweden first year (117 MEUR second year). I assume MEUR= million euros? Define at first use and use acronyms consistently.

Thank you for pointing this out, these paragraphs have been adjusted and the acronyms are written out also in the abstract.

Conclusions: is the high cost of stroke in terms of productivity loss a key finding as well?

Yes, this finding had initially been left out due to word limits but has now been included as well.

Methods:

Is there 100% success in linkage between data sources? If not, note how many were excluded at each step, including decedents?

All individuals that were identified as “true” stroke cases (due to presence in the regional healthcare register as well as in the national quality register of stroke, Swedish Stroke Register) were also identified in the national register containing information related to the social insurance (Social Insurance Agency).

“sick leave can formally be granted to an extent of 25%, 50%, 75% or 100% per day” What does this mean, is this saying that individuals can receive a different % of salary?

It means that the Social Insurance Agency can grant an individual (based on a medical examination performed by the individual’s doctor) formal right to sick leave from work to a given degree. The degree of sick leave is either 25% (sick leave 25% of the working hours, working the remaining 75%), 50%, 75% or 100% (full sick leave, no work). The salary is adjusted accordingly, and the residual is to varying extent (up to a pre-defined ceiling) financed by the government.

The above is now articulated in the manuscript (first paragraph of “Variable definitions”).

The formula for calculating indirect costs could be clarified by giving the formula. Were the adjusted days or raw days used to calculate productivity loss?

The formula is now given under the third paragraph of “Variable definitions” (*[individual income per day] * [individual net days of work absence] * [1.3142]*).

The crude or raw number of net days were used to calculate the productivity loss, i.e. the actual level of productivity loss disregarding the underlying reasons for sick leave or disability pension.

The organization of the methods section could also help the reader by having 2 distinct sections: 1) calculation of indirect costs and 2) modeling outcomes, predictors, and analysis. Currently, it is not clear that these are independent analyses (assuming they are).

Thank you for pointing this out, you are indeed correct – these analyses are independent from each other. The methods section has now been adjusted to better reflect that with the two sub-headings “Calculation of indirect costs” and “Statistical analysis of factors associated to work absence” respectively.

“Participants deceased during follow-up were excluded from the analysis.” Does this mean anyone who died within 2 years? How many were excluded and how might this impact the estimation of indirect costs to society?

Individuals who died within one year after stroke were excluded from the multivariate regression analysis of factors associated with work absence one year after stroke, in order to minimize confounding. However, calculations of indirect costs included all individuals and their registered days of work absence during the follow-up time, to capture as much of the true costs as possible. We have now clarified this with an updated structure of the methods section.

Why did the authors choose to exclude NIHSS from the analysis rather than use multiple imputation approaches? Related, why was mRS not included in the models or descriptive tables?

The frequency of NIHSS reported was very low (48%), and the research group decided, at an early stage, not to perform imputations due to the high likelihood that the sample with reported NIHSS values may be misrepresentative of the cohort.

It is an interesting idea to include mRS as a predictor of costs. We assessed this further in another study within the same research project; *Relationship between functional disability and costs one and two years post stroke*. Lekander I, Willers C, von Euler M, Lilja M, Sunnerhagen KS, Pessah-Rasmussen H, Borgström F. *PLoS One*. 2017 Apr 6;12(4):e0174861. doi: 10.1371/journal.pone.0174861. eCollection 2017. PMID: 28384164.

Please define specification and data source for all covariates in the methods (e.g. living situation, marital status, education etc are listed in the tables but not the methods text)

The covariates and their data sources are now detailed in the text, second paragraph of “Variable definitions”.

Note the recommendations for calculations of productivity loss changed between the first and second panel on cost effectiveness; and this should be cited.

We regret to say that we have not fully understood this comment and what it refers to. We would very much appreciate a clarification.

“The authors did however not have any information whether the individual participant was in fact employed at the time of stroke.” I am confused by this since the intro states that only employed persons whose employers pay into the system can access the health insurance. Could the authors explain how unemployed persons would be included and how this would affect the analysis?

Individuals who 1) work in Sweden or 2) are not working but living in Sweden, are eligible to receive sick leave and/or disability pension. The sentence referred to hence states that the individuals benefitting funding from the government may or may not be employed at the time of stroke. No matter if the individual afflicted with stroke was employed or unemployed at the time of stroke, that individual was, in the health economic perspective, in fact to be considered a potential resource for the labor market.

Model choice seems appropriate for the outcome of “net days”, how were standard errors adjusted for clustering of outcomes by region? How was time accounted for in the model? Also, were two models run: one for year 1 and one for year 2?

This is an interesting idea and would be highly relevant if the outcome was set on regional level and hence could be expected to vary between regions. There was however no adjustment made for clustering of the outcome (net sick leave) by region as the decision is centralized nationally and the national authority (the Social Insurance Agency).

The multivariate regression analysis only treated work absence during the first year after stroke. As deceased individuals were excluded from the regression analysis, follow-up time (365 days) was complete for all included and hence time was not accounted for in the model.

Results:

Per stroke case results from the abstract does not appear in the results, and the discussion numbers do not match either results or abstract. Please improve consistency in which results are reported and confirm accuracy between abstract and manuscript. “The burden of stroke in terms of indirect monetary costs amounted to approximately 17.4 kEUR per stroke case first year (12.2 kEUR second year), totaling approximately 169 MEUR in Sweden first year (117 MEUR second year).”

Thank you for pointing this out. These numbers have been confirmed and the across-age-group averages in the abstract is now also presented in the results section.

Discussion:

“Assuming the equivalent incidence, and equivalent levels of income for the rest of Sweden renders an estimate of approximately 169 Million Euros annually – including only the first year after stroke.” This extrapolation requires some supporting data/rationale in the paper. Are these reasonable assumptions? E.g., if the included sample is limited to people who lived for 2 years does this generalize to all stroke patients nationally? Also, consider moving this finding to the results section, and discuss the assumptions and calculations for this analysis in the methods.

This extrapolation may be considered speculative, and therefore the phrasing of these results has been revised. The rationale behind the extrapolation lies in the fact that the study population is deemed representative with approximately 65% of the Swedish population included and both urban and rural areas represented, and with a representative distribution between age groups.

How do the current estimates compare with prior estimates? Please expand this discussion and report actual indirect costs estimated from citations 13-15 so the reader can compare. Also, would be helpful to state the direct costs of stroke as cited, for the reader to compare.

Thank you for pointing this out, these numbers are now articulated. Furthermore, estimates from additional sources have been added.

The bulleted strengths and limitations of the paper should be included in the discussion text.

Thank you for pointing this out, we have now made sure that they are all included within the discussion section.

Since the conclusion highlights the socio economic gradient, it would be helpful to expand the paragraph explaining finding and potential drivers. Equity is mentioned in the last sentence of the conclusion, and it would be helpful to explain concerns about equity earlier in the discussion.

Thank you for highlighting this, the last paragraph of the discussion section has been expanded consequently. As health equity is a topic treated in the Swedish healthcare act it is relevant to relate the findings regarding distribution of governmental funding to whether it is equitable or not. Disposable income is an example of a factor that should not be associated to social insurance utilization after consideration has been taken to medical factors.

Tables: Disability pension the year prior to stroke (net days) – is the difference between age group is reflecting the larger denominator? If this is the case, consider presenting a per capita rate? The other “net day” items like sick days and inpatient care are similar across age groups, so just wanted to check this one.

Thank you for checking – these numbers are all averages/per capita rates, and the disability pension numbers are actually deviating this much. The deviation is probably due to that disability pension is not prescribed as often to younger individuals with potentially longer work life ahead, and consequently it is relatively more common and overall higher in older age groups.

Living situation – how is this variable defined, is this after or before stroke? Also, why do some variables in Table 2 not appear in Table 1?

Living situation refers to whether the individual lived at home, with or without external support, prior to the stroke. “Stroke severity” and “level of consciousness” refer to the same information and this has been adjusted and now termed only “level of consciousness at hospital arrival”.

In Table 2 only the covariates used in the multivariate regression analysis are included. This fact has now been clarified in the figure caption.

Why are there 3 categories for income in table 1 but 4 in table 2?

The three income values per age group presented in Table 1 refer to values from quartiles 1-3 (25th percentile, 50th percentile and 75th percentile respectively), whilst all four quartiles are included in Table 2 in order to show which quartile is used as reference in the regression analysis.

Figures: something was wrong with the formatting and I could not evaluate these. The axes and labels were not visible on a black background?

Thank you for highlighting this. The figures have now been reproduced.

Reviewer: 2

Comments to the Author

Ischemic stroke and health insurance utilization in a system of universal healthcare and social insurance. Journal: BMJ Open, Manuscript ID bmjopen-2020-043826 Article Type: Original research

The aim of the present study was to assess the population in working age being afflicted with ischemic stroke in a system of universal healthcare and social insurance. By this way the absence from work and related indirect costs were evaluated within the context of the patient's characteristics.

Based on registry data collected in seven Swedish regions between 2008 and 2011 net days of sick leave and disability, indirect costs due to productivity loss and associated factors potentially affecting these social consequences have been assessed.

Compared to men women had significantly fewer net days of sick leave and disability, whereas the high-income group had higher levels of sick leave than low-income groups. There were no differences with regard to educational level, region of birth or civil status.

The indirect costs amounted to approximately 17.400 EUR per stroke case in the first year and 12.200 EUR in the second year. Furthermore the data indicate a socioeconomic gradient in the utilization of the Swedish social insurance.

Informal care has not been evaluated in this study due to lack of data.

General Comments:

- The data sources are somewhat dated, thereby potentially not reflecting the actual situation in Sweden, especially taking into account changes in the Swedish health insurance after the time period under investigation.

We are aware that the data are somewhat dated, and thank you for raising concerns on how the study reflects today's situation. It is however important for us to emphasize that the changes made regarding the health insurance were performed already in 2008 and hence our analysis takes the updated protocols of the health insurance system into account.

- The methodological approach, especially with respect to data acquisition and interactions between different data sources, are not clearly reported (see below).

Thank you for pointing this out. We have now tried to clarify the data sources and their contributions in the methods section. In addition the methods section now has separate sub-headings regarding calculations of indirect costs and regression analysis respectively.

- The presented results have not been compared with data from other industrial countries, which would be of high interest. I also would expect a more detailed comparison of the presented data with other recent publications in this field.

Thank you for pointing this out. We have now articulated and added additional data points in the discussion section to enable the reader to compare with previous findings and put our study into a more nuanced perspective. Information on monetary productivity loss per year due to stroke in other countries is scarce, as shown in previous literature reviews, but significantly lower levels than what was found in this study has been reported (e.g. £ 2 175 for UK). Indirect costs related to stroke incidence in Sweden have been assessed in previous studies and found to be a minor proportion of

the total cost, amounting to 1.99-5.94 kEUR first year. The estimate of total indirect costs of stroke in Sweden in this study (169 MEUR and 117 MEUR respectively) is on par with a 2020 published estimate of 122 MEUR (productivity loss due to morbidity).

- The potential role of primary and secondary prevention has not been discussed.

These are very important matters with regards to the healthcare systems' management of stroke and public health in general. We have now added a comment on this subject at the end of the discussion section although without going into detail as this is somewhat out of the scope for the study.

Specific Comments:

- Data sources and their linkage need to be described in more detail (e.g. Swedish stroke registry, Swedish Social Insurance Agency and socioeconomics and mortality by Sweden statistics). Which data have been acquired from which source? At best a flow chart should be delivered to better understand data sources and acquisition. The Swedish stroke registry needs to be described in detail (primary aim, population, time period, data acquisition and general outcomes).

Thank you for pointing this out, we have now made updates with continuous consideration to word limits. Some more detail is now given under "Variable definitions", with reference to sources leveraged, including variables from the Swedish stroke registry. The aim and coverage rate of the registry are conveyed under "Study population and data sources".

- Baseline characteristics in Table 1 are incomplete with respect to the medical situation (e.g. a complete spectrum of the most important comorbidities, risk diseases apart from atrial fibrillation and hypertension, severity of stroke, medication etc.) and also should include the total numbers of patients under investigation. At best one additional column should deliver the data from the whole population. The levels of consciousness should be defined and definitions should be given at the bottom line of the table. Also other terms like "disposable income" should be defined within the bottom line. Defining the educational level by "years of education" seems to be very vague and does not meet common standards.

Thank you, the numbers of patients under investigation have now been highlighted and inserted at the top of Table 1, and an additional column has been added to present the total numbers.

Instead of presenting particular diagnoses, the variable "Inpatient care year prior to stroke" is presented in Table 1 as it is used as proxy for general morbidity in the studied population (also used in previous publications for similar cohorts, please see examples below) and it is supposed to capture the more severe healthcare need of the study population i.e. the conditions that have implied a need for inpatient care.

Notes have been added beneath the table to describe the variables mentioned in more detail. The categories for educational level are equivalent to high school, secondary school and post-secondary education in Sweden and this is now also conveyed in the table note.

The Association of Pre-stroke Psychosis and Post-stroke Levels of Health, Resource Utilization, and Care Process: A Register-Based Study. Willers C, Sunnerhagen KS, Lekander I, von Euler M. Front Neurol. 2018 Dec 3;9:1042. doi: 10.3389/fneur.2018.01042. eCollection 2018. PMID: 30559711

Sex as predictor for achieved health outcomes and received care in ischemic stroke and intracerebral hemorrhage: a register-based study. Willers C, Lekander I, Ekstrand E, Lilja M, Pessah-Rasmussen H, Sunnerhagen KS, von Euler M. Biol Sex Differ. 2018 Mar 7;9(1):11. doi: 10.1186/s13293-018-0170-1. PMID: 29514685

Hospital comparison of stroke care in Sweden: a register-based study. Lekander I, Willers C, Ekstrand E, von Euler M, Fagervall-Ytting B, Henricson L, Kostulas K, Lilja M, Sunnerhagen KS, Teichert J, Pessah-Rasmussen H. *BMJ Open*. 2017 Sep 7;7(9):e015244. doi: 10.1136/bmjopen-2016-015244. PMID: 28882906

- **Drivers of post-stroke health insurance utilization in Table 2 are not readily comprehensible for the reader and need to be explained at a bottom line or a table legend.**

Thank you, the caption has now been revised to better explain the contents of the table.

- **The Figures all do have a black background and included text passages, numbers or symbols are not readable. This may be the result of a printing problems, but must be resolved.**

Thank you for highlighting this. The figures have now been reproduced.

- **Results: The text passage “The loss of productivity, calculated based on individual participants’ income together with additional employer contributions, amounted to 110 Million Euros during the first year after stroke (Figure 2)” is unclear. Do the 110 Million Euros refer to the whole population under investigation? How do these data compare with other European countries?**

Yes, the 110 MEUR refer to the whole population under investigation, i.e. the approximately 65% of the Swedish population. The sentence has been revised to be more concise. These costs are on par with other estimates (e.g. Luengo-Fernandez 2020, reference 19) but vary significantly compared to other countries (e.g. less than half of Norway’s productivity loss whilst Norway is a similar but significantly less populated country, but very similar to total productivity loss of the Netherlands).

- **Results, page 10, lines 26-29: “When adjusting for other relevant factors of potential association with levels of absence after stroke, belonging to a certain age group was not significantly associated to differences in absence from work due to sick leave or disability pension”. Which “other relevant factors” have been included into the calculations?**

The other relevant factors referred to are the other covariates included in the multivariate regression analysis, and this sentence has now been rephrased.

- **Page 14, line 6: “During 2008, significant changes were implemented in the Swedish health insurance to limit the length of time possible to receive support for loss of income due to disability”. It may be regarded as a relevant limitation of the study by evaluating a dated study population (see general comment above).**

It is a valid point that the study population may seem dated. The updated protocols for the Swedish health insurance, as referred to, were however already in place for this study population, starting in 2008.

- **Page 15, line 24: “Individuals that are afflicted with stroke in working age have a significantly higher pre-stroke level of health insurance utilization” This comment**

underscores the relevance of risk factors and comorbidities of stroke patients, potentially being present decades before incident stroke. Primary prevention therefore is essential for the individual but also for the community supporting the individual. This has not sufficiently been addressed neither within the study nor within the manuscript.

Thank you for pointing this out, this is a very interesting and highly relevant perspective. Even after adjusting for e.g. previous social insurance utilization, the differences in post-stroke utilization are significant between income groups. Probably there is a socioeconomic gradient also regarding the knowledge of and ability to pursue primary and secondary prevention. Even though this is not part of the main scope of the study we have therefore added a comment at the end of the discussion section.

VERSION 2 – REVIEW

REVIEWER	Brystana Kaufman Duke University, USA
REVIEW RETURNED	15-Jan-2021

GENERAL COMMENTS	The revised manuscript is much clearer in the description of methods, and the results figures are now legible. While my prior comments were mostly addressed, this new information has raised additional questions that should be addressed prior to publication. Key concerns: First, the explanation for the higher proportions of female stroke in younger age groups is inaccurate and must be corrected. Strongly recommend changing this explanation to reflect the cited evidence, for example restating the key points from the cited work to avoid misrepresentation. Second, the results of the multivariable regression need to be presented in a form that is interpretable, for example incidence rate ratios. More detail in comments below. Prior comment: Note the recommendations for calculations of productivity loss changed between the first and second panel on cost effectiveness; and this should be cited. We regret to say that we have not fully understood this comment and what it refers to. We would very much appreciate a clarification. To clarify, here is the citation to the second panel on cost-effectiveness in health and Medicine. Authors should follow and cite these best practices in calculating productivity loss to support use of their results in cost-effectiveness analyses. https://jamanetwork.com/journals/jama/article-abstract/2552214 New Comments: Abstract: There wasn't a "multivariate" analysis, do you mean multivariable? Methods – page 6, statistical analysis: "health insurance utilization" can be measured many ways. please clearly state the dependent variable of the regression model and specification. Results: The magnitude and CI for key significant associations from the negative binomial model should be presented in the text. I strongly recommend presenting and discussing incidence rate ratios. The interpretations of raw coefficients of a negative binomial model are challenging (change in log of expected count). The new sections and headings in the methods work well; please do the same for results section. Flipping the order of presenting the Figure 1 and 2 results would be helpful to the reader so all the health utilization results are together, and the productivity results at the end – again demonstrating these are separate analyses. Discussion: The first paragraph is very long. Avoid repeating information already in the results. What are the key points that will be discussed? "Higher proportion of women in younger age categories may be
--

	related to impacted hormone levels from contraceptives or pregnancy, e.g. changed balance in coagulation factors or hypertensive crisis” This statement does not reflect the conclusions of the cited work. In the cited work, the key points clearly state “Hypertension and atrial fibrillation, key risk factors for stroke, are more frequent in women than in men” while risk due to hormones and contraceptives is very small (and not even mentioned in the key points). “Absence the year prior to stroke was positively correlated with absence the year after stroke. Interestingly, when this baseline characteristic was not taken into account, several socioeconomic indicators showed significant association with absence the year after stroke, including educational level, birth region and civil status.” This should be presented as a sensitivity analyses (consider including table as supplemental material) and reported in the results. “Individuals that are afflicted with stroke in working age have a significantly higher pre-stroke level of health insurance (sick leave or disability pension) utilization than the average in Sweden.” What is the average in Sweden? Key limitations that need to be explicitly stated: Data are now 10+ years old, discuss any current trends that may impact estimates; measurement error - lack of information on current employment at time of stroke Table 1: labels are still unclear. Acronyms need to be defined within the table or footnotes (e.g. ADL, Q1, Q2, Q3). If 25, 50 and 75 percentile state this, and, consider presenting as median (IQR xx-xx). Best practice is to include a measure of variation (e.g. standard deviation) for continuous measures (inpatient days, sick leave, disability days). Table 2: Not sure about the journal style, but I expect the information currently in the title would be better as a footnote. Consider title such as “Associations between patient and stroke characteristics with net days of absence following stroke” Figure 1: define acronyms (IS) and add confidence intervals
--	--

REVIEWER	Prof. Dr. med. Bernhard Rauch FESC Institut für Herzinfarktforschung - IHF, Ludwigshafen, Germany
REVIEW RETURNED	03-Jan-2021

GENERAL COMMENTS	Health insurance utilization after ischemic stroke in Sweden: a retrospective cohort study in a system of universal healthcare and social insurance Journal: BMJ Open, Manuscript ID bmjopen-2020-043826 R1 Article Type: Original research Evaluation of the revised manuscript (revision 1) (Prior heading: “Ischemic stroke and health insurance utilization in a system of universal healthcare and social insurance”. Journal: BMJ Open, Manuscript ID bmjopen-2020-043826 Article Type: Original research) General comments: - The presented data are important and of general interest - the authors have addressed the comments of the reviewers, but there are still some shortcomings, which need to be addressed. Specific major comments: - data sources still are not sufficiently transparent. Please summarize the data sources in a table at best with a data link or
---

	reference.  - the role of primary and secondary prevention of stroke is essential for both, the individuals and the socio-economic system by significantly influencing general costs and sick leave. This still has not been sufficiently pointed out in the discussion. Moreover, this still needs to be set in a context to what we know from other countries with respect to ischemic cardiovascular disease – at least as far as this is possible on the basis of the actual literature (e.g. Kotseva et al EJPC 2019; Bejot et al 2016; Lecoffre C et al. Stroke 2017, Jennum P et al BMC Health Serv Res 2015 and others) - The discussion needs to be completed by clearly pointing out the “strengths” and the “limitations” of the study. Specific minor comments:  - Statistical analysis...: please consider the English language, e.g. “select factors”, “presented under Results...” - Figure 1: the numbers of the ordinate and abszissa need to be supplemented with a dimension (e.g. “days” – y-axis; “years of age” – x-axis). This would help the reader. - The list of abbreviations should be completed, e.g. “MEUR” and “kEURO” - Page 33 of 40, paragraph 4, comment referring to Fig. 2 should be checked for English language
--	--

VERSION 2 – AUTHOR RESPONSE

Reviewer: 2

General comments:

- The presented data are important and of general interest
- the authors have addressed the comments of the reviewers, but there are still some shortcomings, which need to be addressed.

Specific major comments:

- data sources still are not sufficiently transparent. Please summarize the data sources in a table at best with a data link or reference.

Thank you for stressing this. Table 1 with data on the study population has been updated to include a column in which the data source is specified. This is also referenced within the first paragraph of the methods section.

- the role of primary and secondary prevention of stroke is essential for both, the individuals and the socio-economic system by significantly influencing general costs and sick leave. This still has not been sufficiently pointed out in the discussion. Moreover, this still needs to be set in a context to what we know from other countries with respect to ischemic cardiovascular disease – at least as far as this is possible on the basis of the actual literature (e.g. Kotseva et al EJPC 2019; Bejot et al 2016; Lecoffre C et al. Stroke 2017, Jennum P et al BMC Health Serv Res 2015 and others)

We agree that the discussion on prevention is a very important topic relating to the burden of stroke. Therefore, the discussion has been extended to better cover this perspective. This part of the discussion has however been kept somewhat limited as the focus of the study was utilization of the health insurance. The following text now ends the last paragraph of the discussion section:

“It is also likely that underlying themes such as knowledge of and ability to pursue primary and secondary stroke prevention is asymmetrically distributed across e.g. income groups. Such patterns reinforces already existing systematic differences in ill-health, and actions to impact that are key to change the situation.”

Thank you for recommendations on previous research to put the present study into context.

Benchmarks from previous research have been added within the first paragraph of the discussion section.

- The discussion needs to be completed by clearly pointing out the “strengths” and the “limitations” of the study.

Thank you for pointing this out. These parts of the discussion have been revised and now more clearly describes the possible strengths and limitations of the study. The updated text now reads: “There are several limitations with this study. Informal care is costly, and no estimation of informal care has been included in this study due to lack of reliable data. Furthermore, the actual costs measured are also in fact higher as any cases of sick benefits lasting shorter than 14 days are not registered in the Social Insurance Agency database. There is also a lack of information whether the individuals were employed at the time of stroke or not, which would have been valuable to nuance the perspective on health insurance utilization versus unemployment. The data may also be regarded as old as the first data point is more than 10 years old at the time of publication.

There are also several strengths with the present study. The underlying data are to a large extent extracted from population registries which implies very high coverage rates and hence a close-to-true picture of the actual population, and sick-leave costs for stroke patients have been calculated before as well as after the stroke event, which adds important information in terms of incremental costs after stroke. The possibility to adjust for several factors such as sex, age, comorbidity as well as several sociodemographic factors enables a better understanding of what patient characteristics that are in fact associated with health insurance utilization. The computation of productivity loss is based on objective data points for registered work absence and not related to any preference-based measures for health measures. Therefore, the findings regarding indirect costs related to stroke from the present study could be used together with preference-based measures and estimations of impact on health-related quality of life from other studies without implying risk for double counting, in line with previously published recommendations [24].”

Specific minor comments:

- Statistical analysis...: please consider the English language, e.g. “select factors”, “presented under Results...”

This sentence has now been updated and the wording “patient characteristics” is used.

- Figure 1: the numbers of the ordinate and abszissa need to be supplemented with a dimension (e.g. “days” – y-axis; “years of age” – x-axis). This would help the reader.

The figure title has now been adjusted to include a description of the axes of the graph.

- The list of abbreviations should be completed, e.g. “MEUR” and “kEURO”

Thank you for noticing this, the list of abbreviations has been extended to include the above and should now be considered complete.

- Page 33 of 40, paragraph 4, comment referring to Fig. 2 should be checked for English language

Thank you, the last sentence of the paragraph has been adjusted (first paragraph of the presentation of results relating to Figure 2).

Reviewer: 1

Comments to the Author:

The revised manuscript is much clearer in the description of methods, and the results figures are now legible. While my prior comments were mostly addressed, this new information has raised additional questions that should be addressed prior to publication.

Key concerns: First, the explanation for the higher proportions of female stroke in younger age groups

is inaccurate and must be corrected. Strongly recommend changing this explanation to reflect the cited evidence, for example restating the key points from the cited work to avoid misrepresentation. Second, the results of the multivariable regression need to be presented in a form that is interpretable, for example incidence rate ratios. More detail in comments below.

Thank you for your comments.

We have adjusted the phrasing regarding stroke in women of younger age to include all main factors affecting the stroke risk that were reported in the referenced research.

The presentation of coefficients has been kept but statistical significance is referenced in the text in the results section, and the results from the sensitivity analysis have been appended.

Prior comment: Note the recommendations for calculations of productivity loss changed between the first and second panel on cost effectiveness; and this should be cited. We regret to say that we have not fully understood this comment and what it refers to. We would very much appreciate a clarification. To clarify, here is the citation to the second panel on cost-effectiveness in health and Medicine. Authors should follow and cite these best practices in calculating productivity loss to support use of their results in cost-effectiveness analyses.

<https://eur01.safelinks.protection.outlook.com/?url=https%3A%2F%2Fjamanetwork.com%2Fjournals%2Fjama%2Farticle-abstract%2F2552214&data=04%7C01%7Ccarl.willers%40ki.se%7C5126e3b5758d45d7a8ab08d8bbd1b408%7Cbff7eef1cf4b4f32be3da1dda043c05d%7C0%7C0%7C637465857384543818%7CUnknown%7CTWFpbGZsb3d8eyJWIjoiMC4wLjAwMDAiLCJQIjoiV2luMzliLCJBTiI6IjEhaWwiLCJXVCi6Mn0%3D%7C3000&sdata=T9I5Jn96VFyapyLdl0e4OAnJB%2BnnMq2SGXvRv7Pz2V4%3D&reserved=0>

Thank you for clarifying this.

We have now incorporated a reference to the second panel on cost effectiveness and added an explicit statement on the methods for computation on productivity loss:

“The computation of productivity loss is based on objective data points for registered work absence and not related to any preference-based measures for health measures. Therefore, the findings regarding indirect costs related to stroke from the present study could be used together with preference-based measures and estimations of impact on health-related quality of life from other studies without implying risk for double counting, in line with previously published recommendations [23].“

New Comments:

Abstract: There wasn't a "multivariate" analysis, do you mean multivariable?

Thank you for noticing. The term "multivariable" is now used consequently throughout the manuscript.

Methods – page 6, statistical analysis: "health insurance utilization" can be measured many ways. please clearly state the dependent variable of the regression model and specification.

This paragraph has been updated to clarify this, and the text now reads:

“Health insurance utilization was analyzed; net days of sick leave and disability pension (early retirement due to disability) were used as dependent variable, treated as count variable, and analyzed with negative binomial regression. The reason for treating it as a count variable was that the decision on the total amount of sick leave is iterative and based on a joint agreement between the doctor and the patient done during several occasions.”

Results: The magnitude and CI for key significant associations from the negative binomial model should be presented in the text. I strongly recommend presenting and discussing incidence rate ratios. The interpretations of raw coefficients of a negative binomial model are challenging (change in log of expected count).

The associations discussed in the text are now presented in the text with regards to statistically significant association.

The new sections and headings in the methods work well; please do the same for results section. Flipping the order of presenting the Figure 1 and 2 results would be helpful to the reader so all the health utilization results are together, and the productivity results at the end – again demonstrating these are separate analyses.

Thank you for noticing this, the results section has now been updated accordingly.

The order of the figures has been kept as Figure 1 presents the levels of work absence in terms of net days and Figure 2 illustrates the indirect costs which were computed based on work absence numbers and individual levels of income, i.e. we look at it as though Figure 2 in that sense is based on data from Figure 1. Hopefully you can see why we would choose to keep this order.

Discussion: The first paragraph is very long. Avoid repeating information already in the results. What are the key points that will be discussed?

The first paragraph has been revised and now reads;

“The burden associated with stroke is immense – in terms of Health-Related Quality of Life [1] as well as money [13-15]. In addition to severe personal suffering, stroke affects society by putting a significant burden on direct healthcare resources but also by indirect, major and long-lasting productivity loss. The first-year indirect monetary costs for stroke in the present study totaled 110 Million Euros. Given that almost two thirds of the Swedish stroke cases were the basis for analysis and that both urban and rural areas are represented, it is likely not unfair to make an extrapolation for all of Sweden; assuming equivalent incidence and equivalent levels of income for all of Sweden renders an estimate of approximately 169 Million Euros annually – including only the first year after stroke. The costs are not limited to the first or second year (indirect monetary costs of 76 MEUR for the study population, estimated to 117 MEUR for the full Swedish population), but expected to last several years after the stroke event [16]. Information on monetary productivity loss per year due to stroke in other countries is scarce, as shown in previous literature review [17], but significantly lower levels than what was found in this study has been reported (e.g. £ 2 175 per case for UK [18]). A study from Denmark, a context similar to the present study, estimated the indirect costs to 3.1 kEUR, although the majority of the subjects was no longer working (58.2% above 70 years of age) [19]. Indirect costs related to stroke incidence in Sweden have been assessed in previous studies and found to be a minor proportion of the total cost, amounting to 1.99-5.94 kEUR per stroke first year [13-15]. These studies did however include persons of all ages afflicted with stroke, and likely the indirect costs are significantly higher for the group in general working age as included in the present study. The estimate of total indirect costs of stroke in Sweden in this study is on par with a 2020 published estimate of 122 MEUR (productivity loss due to morbidity) [20].”

“Higher proportion of women in younger age categories may be related to impacted hormone levels from contraceptives or pregnancy, e.g. changed balance in coagulation factors or hypertensive crisis” This statement does not reflect the conclusions of the cited work. In the cited work, the key points clearly state “Hypertension and atrial fibrillation, key risk factors for stroke, are more frequent in women than in men” while risk due to hormones and contraceptives is very small (and not even mentioned in the key points).

Thank you for commenting on this finding.

All main reasons reported in the referenced research are now mentioned within this paragraph to better nuance the discussion;

“The proportion of women over the different age groups differed; the younger the age group, the higher the proportion of women. The reasons for this are several; hypertension and atrial fibrillation are less frequent in men than in women, whilst risk factors such as atrial fibrillation are of higher impact in women. In addition, pregnancy and contraceptives affect hormone levels and increase the risk for stroke [22].”

“Absence the year prior to stroke was positively correlated with absence the year after stroke. Interestingly, when this baseline characteristic was not taken into account, several socioeconomic indicators showed significant association with absence the year after stroke, including educational level, birth region and civil status.” This should be presented as a sensitivity analyses (consider including table as supplemental material) and reported in the results.
Thank you for pointing this out. A table is now appended as supplementary material to present these results, and this sensitivity analysis is reported under the results section.

“Individuals that are afflicted with stroke in working age have a significantly higher pre-stroke level of health insurance (sick leave or disability pension) utilization than the average in Sweden.” What is the average in Sweden?

This is a relevant question, and we have chosen to discuss this under the second paragraph in the discussion section; the average in Sweden amounted to 6.0 net days at the time whilst the study population average amounted to almost 18 net days.

Key limitations that need to be explicitly stated: Data are now 10+ years old, discuss any current trends that may impact estimates; measurement error - lack of information on current employment at time of stroke

Thank you for pointing this out. The discussion on limitations has now been extended and more clearly structured.

Table 1: labels are still unclear. Acronyms need to be defined within the table or footnotes (e.g. ADL, Q1, Q2, Q3). If 25, 50 and 75 percentile state this, and, consider presenting as median (IQR xx-xx). Best practice is to include a measure of variation (e.g. standard deviation) for continuous measures (inpatient days, sick leave, disability days).

Thank you for noticing this, all acronyms are now specified within the footnote.

Updates have also been made regarding the presentation of median and IQR instead of quartiles. Standard deviations for continuous variables have been added to the table.

Table 2: Not sure about the journal style, but I expect the information currently in the title would be better as a footnote. Consider title such as “Associations between patient and stroke characteristics with net days of absence following stroke”

Thank you for noticing this, the figure title has been updated.

Figure 1: define acronyms (IS) and add confidence intervals

The figure title has been adjusted to include this acronym. Confidence intervals have been added in a table appended as supplementary material (Table A2).

VERSION 3 – REVIEW

REVIEWER	Brystana Kaufman Duke University United States
REVIEW RETURNED	02-Mar-2021

GENERAL COMMENTS	My comments were largely addressed, with the exception of the request to present IRR rather than coefficients. While the results presented are not technically inaccurate, they are difficult to interpret.
---

REVIEWER	Prof. Dr. med. Bernhard Rauch FESC IHF - Institut für Herzinfarktforschung, Ludwigshafen, Germany
REVIEW RETURNED	23-Feb-2021

GENERAL COMMENTS	the manuscript has been approved according to the recent suggestions. I still think that there should be some polishing of the English language, which probably will professionally be done by the Publisher. Taking together, the presented data are of a high socioeconomic relevance.
---